# Low Temperature Adhesive Bonding-Based Fabrication of an Air-Borne Flexible Piezoelectric Micromachined Ultrasonic Transducer

**DOI:** 10.3390/s20113333

**Published:** 2020-06-11

**Authors:** Wei Liu, Dawei Wu

**Affiliations:** State Key Laboratory of Mechanics and Control of Mechanical Structures, Nanjing University of Aeronautics and Astronautics, Nanjing 210016, China; liuw2288@nuaa.edu.cn

**Keywords:** flexible, piezoelectric micromachined ultrasonic transducer (PMUT), air, low temperature adhesive bonding, time of flight (TOF)

## Abstract

This paper presents the development of a flexible piezoelectric micromachined ultrasonic transducer (PMUT) that can conform to flat, concave, and convex surfaces and work in air. The PMUT consists of an Ag-coated polyvinylidene fluoride (PVDF) film mounted onto a laser-manipulated polymer substrate. A low temperature (<100 °C) adhesive bonding technique is adopted in the fabrication process. Finite element analysis (FEA) is implemented to confirm the capability of predicting the resonant frequency of composite diaphragms and optimizing the device. The manufactured PMUT exhibits a center frequency of 198 kHz with a wide operational bandwidth. Its acoustic performance is demonstrated by transmitting and receiving ultrasound in air on curved surface. The conclusions from this study indicate the proposed PMUT has great potential in ultrasonic and wearable devices applications.

## 1. Introduction

Ultrasound has been widely used in non-destructive testing (NDT) [1,2], medical diagnostics and therapy [3,4,5], and sensing detection [6,7,8] because of its exceptional features such as noninvasiveness, convenience, high penetrability and sensitivity. Ultrasonic transducers, which are key components of any ultrasound system, are either configured with inflexible structures or fabricated from bulk piezoelectric materials. These rigid architectures afford stable performance and favorable piezoelectric properties but prevent ultrasonic transducers from being used on irregular nonplanar surfaces which widely exist in real objects.

Recent advances in flexible electronics provide innovative materials and fabrication processes making it possible to realize flexible ultrasound devices that can be coupled with nonplanar surfaces [9,10,11,12,13,14,15]. For example, piezoelectric nanofibers with excellent properties are one of the types of materials proposed for use in wearable electronics [16,17,18,19], and some sensors can be easily embedded as a part of human skin or clothing for health monitoring by the near field electrospinning (NFES) technique [20,21,22]. Among these devices, flexible piezoelectric micro-ultrasonic transducers have advantages over traditional rigid ultrasonic transducers in terms of weight, volume, adaptability and portability. Available feasible strategies for fabricating flexible piezoelectric micro-ultrasonic transducers that can be mainly divided into two categories: island-bridge connection techniques and transfer printing techniques. In the former case, the flexibility is achieved by connecting bulk piezoelectric ceramic islands to each other using polymer joints or embedding piezoelectric ceramics into patterned polymer holes [23,24,25,26,27]. This island-bridge connection technique is based on micromachining fabrication processes, including standard photolithography, deposition and etching, which require expensive equipment, hazardous chemical substances, photomasks, and a clean room ambient. Moreover, the temperatures of some deposition and etching processes are higher than 100 °C, resulting in poor process adaptability and limited material selection. Besides, multilayered electrode designs and complex stack structures doubtlessly increase the cost and the risk of the manufacturing process errors. The transfer printing technique, by contrast, is considered to be a better option to fabricate flexible thin-film ultrasonic transducers. The devices are fabricated on a rigid silicon donor substrate firstly and then transferred onto a receiver substrate using an elastic stamp [28]. In this technique, the control of the interfacial adhesion force between the stamp and the structure is critical to the success of the transfer printing process [29,30]. However, the insufficient strong to weak adhesion force ratio needs an additional sacrificial layer or surface treatment to decrease the interfacial adhesion force between the structure and the donor substrate, which certainly increases the cost and risk of the fabrication process. Although some other methods have also been proposed to fabricate flexible piezoelectric micro-ultrasonic transducers, the limitations of existing approaches still remain, such as large-size dicing process [31], time-consuming deep silicon etching [32], and unreliable cavity structures [33], all of which compromise the stability and convenience of the fabrication process. 

In this paper, we report the development of an air-borne flexible piezoelectric micromachined ultrasonic transducer (PMUT) based on a simple and robust low temperature adhesive bonding technique. A laser precision machining system is applied in this technique to form structures directly and rapidly, and the maximum instantaneous temperature of the whole process is controlled below 100 °C. Finite element analysis (FEA) is made use of to predict the resonant frequency of composite diaphragms and optimize the structure. The resulting device has a center resonant frequency at 198 kHz with good conformal contacting with flat, concave, and convex surfaces, as well as the skin. The acoustic characterization is discussed and the time of flight (TOF) is successfully captured by curved PMUT. The experimental results demonstrate that our PMUT has great potential in wearable ultrasonic sensor applications.

## 2. Design and Modeling

A schematic cross-section of our PMUT structure is shown in Figure 1. It consists of a piezoelectric layer sandwiched between Ag electrodes and bonded on a polymer substrate. A polyvinylidene fluoride (PVDF) film has been selected as the piezoelectric layer, given its unique dielectric and piezoelectric properties, as well as its low impedance, wide bandwidth, high sensitivity, optical transparency, and mechanical flexibility [34,35,36]. The one-side Ag coated commercial PVDF sheet (Zhimeikang Technology Co., Ltd., Shenzhen, China) is divided into square film pieces with 10 mm side length to make them easy for use in the later fabrication process, which is shown in Figure 2a. The thickness of PVDF film is 28 μm measured by a spiral micrometer (Figure 2b), and the total thickness of one-side Ag coated PVDF is 38 μm (Figure 2c). The top Ag electrode is patterned on PVDF film and the diameter is 460 μm. The well-known thermosetting polyimide (PI) has been used as the passive layer and bonded layer simultaneously. The thickness of PI can be precisely controlled by spin coating, and PI also has excellent versatility and machinability. The commercial Kapton film has been served as the substrate to form a suspended structure of composite plates. The sidewall is 750 μm in diameter and the cavity is 100 μm in depth. All of the materials we used are flexible, which leads to favorable flexibility of the whole device.

There are two types of vibration mode in ultrasonic transducers: longitudinal vibration mode (also called thickness vibration mode) and flexural vibration mode. For the former, the resonant frequency of the ultrasonic transducer is directly proportional to the longitudinal wave velocity of the piezoelectric material and is inversely proportional to the thickness of the piezoelectric layer [37]. Once the piezoelectric material has been chosen, the longitudinal wave velocity is immutable consequently, resulting in that the resonant frequency is solely dependent on the thickness, which limits the geometrical dimensions and design flexibility. In comparison, the flexural vibration mode is not directly related to the thickness of piezoelectric materials. Instead, the shape, dimensions, and boundary conditions all affect the resonant frequency, which makes the design more flexible and extensible. For our PMUT, the flexural vibration mode of the circular film is utilized. The PVDF film is polarized in the direction of its thickness, and the natural vibration frequency of a circular film with an edge-fixed boundary condition can be computed from the following equations [38]: (1)$$f=\frac{\lambda_{mn}{}^{2}}{2\pi r^{2}}\sqrt{\frac{D}{\rho h}}$$
(2)$$D=\frac{Eh^{3}}{12\left(1-\nu^{2}\right)}$$
where r is the membrane radius, h is the thickness of the membrane, ρ is the mass density of the membrane, D is flexure rigidity, E is Young’s modulus, ν is Poisson’s ratio, and $\lambda_{mn}$ is the natural frequency constant. The first three natural frequency constants for fixed edge boundary condition are given as follows: $\lambda_{00}=3.196$, $\lambda_{01}=4.611$, $\lambda_{02}=5.906$ [39]. The typical values of the mass density, Young’s modulus, and Poisson’s ratio of PVDF film are 1780 $\mathrm{kg}/m^{3}$, 3 GPa, and 0.29 in sequence [40]. The thickness of PVDF is 28 μm and the membrane diameter is set to 750 μm. Through Equations (1) and (2), we can calculate that the first resonance frequency of PVDF film is 130.2 kHz, which is suitable for air-coupled applications. Realistically, all of the diaphragms including the piezoelectric layer and electrodes have contributed to vibration. In the case of multilayer diaphragms, various material constants should be considered. 

Here, the finite element analysis (FEA) method was also applied to predict the resonant frequency of composite diaphragms more reasonably. A simulation model of the PMUT built using COMSOL Multiphysics 5.3 is shown in Figure 3a. The 2D axisymmetric model and piezoelectric devices multiphysics interface were chosen. The geometry has been built in software by considering all the dimensions and material constants. The thicknesses of the top Ag electrode, PVDF film, bottom Ag electrode, and PI passive layer are 10, 28, 10 and 4 μm, respectively. The air cavity beneath the stack is 750 μm in diameter and the top electrode is optimized at 460 μm in diameter (see the section Optimization of the PMUT for details). For the material properties of PVDF the built-in material library in COMSOL Multiphysics 5.3 is employed and other properties are listed in Table 1. The fixed boundary conditions are applied at the edge of the model. Eigenfrequency analysis is used to predict the natural frequency and mode shapes of the structure. The first flexural mode shape of PMUT is shown in Figure 3b and the absolute value of the admittance is plotted in Figure 3c. From the FEA simulation results, the natural frequency is revealed to lie at approximately 202.58 kHz, which is higher than the theoretical analysis value on account of considering the effect of multilayer diaphragms. According to Equations (1) and (2), when r is fixed, the resonance frequency (f) is proportional to the first power of h. We assume that the membrane thickness has a greater effect on the resonant frequency than other parameters. In the case of multilayer diaphragms, h is defined as the thickness of the whole diaphragm, including top and bottom electrodes, PVDF piezoelectric layer, and PI passive layer, which is bigger than the thickness of a single PVDF layer, resulting in a higher frequency than given by the theoretical calculation. To verify this assumption, an additional FEA for only one 28 μm thick PVDF layer is applied. The dimension size and boundary conditions are the same as before, and the simulation results are shown in Figure 4. The resonant frequency of PVDF based monolayer structure is 126.16 kHz, which agrees with the theoretical calculation approximately. Thus, FEA is competent to predict the resonant frequency of composite diaphragms conveniently.

## 3. Fabrication Process

The low temperature adhesive bonding fabrication process is divided into pre-processing and main processing steps. A laser precision machining system (ProtoLaser U3, LPKF Tianjin Co., Ltd., Tianjin, China) is used to quickly pattern printing masks and flexible substrates in pre-processing. Compared with micromachining fabrication techniques such as standard photolithography, depositing, and etching, laser precision machining is convenient and time-saving. The diameter of focused laser beam is 30 μm, and the minimum space of ultra-fine structures can reach 45 μm, which will meet the requirements of micro devices fabrication not only in design stage but also in massive production stage. Ten μm thick commercial Kapton tapes were subjected to the laser machining system to form printing masks, and 100 μm thick Kapton sheets were also processed by laser to gain flexible substrates. The key points for patterning Kapton polymers are controlling the laser power to completely ablate the pattern while avoiding the contour distortion caused by excessive ablation. The laser parameters have been optimized to process Kapton polymers with the highest yield, which are given in Table 2. Figure 5 shows the results of the laser-machined printing mask (Figure 5a) and flexible substrate (Figure 5b). The diameter of the through-hole on Kapton substrate is 750 μm and four symmetrical triangles placed at the edges were acted as the alignment marks in adhesive bonding process. On the other hand, the temporary carrier consisting of a layer of commercial thermal release tape (TRT, Shunsheng Electronics Co., Ltd., Shenzhen, China) mounted on a glass sheet also has been prepared in advance, which is shown in Figure 6. TRT is a kind of tape with feature of convertible adhesive force and is widely used for transferring graphene [41]. When it is heated to 90~100 °C, the adhesive force will vanish irreversibly. This characteristic of TRT will make it convenient for samples to transfer in fabrication process.

The schematic main processing steps are illustrated in Figure 7. It started with patterning the top Ag electrode on PVDF film. The printing mask was pasted on the PVDF film and the conductive silver paste (0.3 mL, Zhimeikang Technology Co., Ltd., Shenzhen, China) was coated on it evenly using a special PET scraper (Figure 7a). A 30 min baking step at 60 °C was performed for curing conductive silver paste in thermostatic oven. After that, the printing mask was peeled off and the patterned Ag electrode was remained on PVDF film (Figure 7b). Next, the Ag coated PVDF film was pasted on the temporary carrier upside down (Figure 7c). A layer of thermosetting PI (20% solid content, Qiancheng Plasticizing Material Co., Ltd., Dongguan, China) was spun at 3000 RPM onto the PVDF film to obtain a thickness of 4 μm (Figure 7d). To assemble the device, the prepared substrate was aligned and adhesively bonded on the PVDF film (Figure 7e). Finally, the sample was baked at 60 °C for 10 min to solidify the PI layer and then baked at 95 °C for 5~10 s (Figure 7f) to deactivate the TRT and separate the device from the temporary carrier (Figure 7g). 

The processing results of some steps are shown in Figure 8. Figure 8a shows the PVDF film was firstly fixed on the temporary carrier and then a laser processed printing mask was pasted on it. The four symmetrical triangles placed on the printing mask were coincided with the contour of PVDF film for aligning. Then the conductive silver paste was daubed repeatedly to ensure that the pattern of the electrode was copied from printing mask to PVDF completely, which was shown in Figure 8b. Because the adhesive force between PVDF film and temporary carrier is higher than it between PVDF film and printing mask, the printing mask could be peeled off carefully and the PVDF film remained on the temporary carrier. An optical image of the Ag patterned PVDF film is shown in Figure 8c. Figure 8d shows the sample fixed on temporary carrier ready for adhesive bonding. The substrate was picked up by a polydimethylsiloxane (PDMS) film covered glass sheet and then bonded on PVDF film without extra bonding pressure. The final device is shown in Figure 8e. Through optical microscope observation, no residues remained on the PMUT surface after deactivating the TRT. The device is not broken after the separation process according to simple capacitance testing.

## 4. Results and Discussion

### 4.1. Optimization of the PMUT

There are some main parameters of PMUT design, such as the thickness of the PVDF piezoelectric layer and the diameter of the cavity, whose changes will exert a great influence on the resonance frequency. These parameters can be regarded as non-optimizable values in this experiment. Besides, other parameters like the diameter of the Ag top electrode and the thickness of the PI passive layer can be considered as optimizable parameters because tiny changes won’t affect the resonance frequency greatly but could improve the performance of the device. 

First, we optimized the diameter of top Ag electrode to maximize the deflection displacement of composite diaphragms. The radius of the top electrode is used as a variable, and the deflection displacement of composite diaphragms is taken as the criterion. The thicknesses of the top Ag electrode, PVDF layer, bottom Ag electrode, and PI layer were 10, 28, 10 and 4 μm, respectively. The air cavity is also 750 μm in diameter. The top electrode radius ranges from 200 μm to 260 μm with a step size is 10 μm. A measuring point is located at the center of the upper surface of the model to record the deflection displacement in longitudinal direction. Figure 9a shows the deflection displacement of the measuring point under different frequencies. For comparison conveniently, the *y*-axis represents the ratio of the deflection with different radii to maximum deflection. From the chart, when the radius of the top electrode is 230 μm (the diameter is 460 μm), the deflection is maximum, which means the acoustic wave generated by the PMUT has the highest sound pressure level in this case. From the admittance chart shown in Figure 9b, we can also see the resonance frequency increases with the radius. The resonance frequencies of the PMUT with different radii of top electrodes are listed in Table 3. The arithmetic average frequency increase rate is 4.07%, which illustrates that the resonance frequency is affected by the diameter of top Ag electrode to a great extent. 

Then we studied the influence of the PI passive layer on the deflection displacement of composite diaphragms. The role of the passive layer is to support the piezoelectric stack and provide mechanical restoring force at the same time. The thickness of the passive layer ranges from 1 μm to 7 μm with a step size is 1 μm. The diameter of the top electrode is set to 460 μm according to the above optimization result, and the other parameters are maintained. A measurement point is placed at the center of the upper surface to record the longitudinal deflection. Especially, in an extreme case, no passive layer also has been considered. The displacement of the measurement point under various conditions is shown in Figure 10a. From this result, we can obviously see that the deflection has a maximum peak when the thickness of the passive layer is 4 μm. The absolute value of admittance is plotted in Figure 10b. The results show that the resonant frequency first increases and then decreases along with the increasing thickness of the passive layer, the lowest resonant frequency appears when the thickness is 3 μm. In the case of no passive layer, the resonant frequency has a huge shift but the deflection displacement has no obvious change. The arithmetic average frequency increase rate is 0.34%, which means that the thickness of PI passive layer has little effect on the resonant frequency. 

### 4.2. Simulation of the Sound Field

Analyzing the sound field of an ultrasonic device is important for its applications. The acoustic-piezoelectric interaction, frequency domain multiphysics interface was adopted in COMSOL 5.3. In order to improve computation efficiency, a simulation model with 2D axial symmetry was established. A cylindrical air domain with a radius of 2 mm and a height of 3 mm was placed in front of the PMUT model. Perfectly matched layers (PMLs) were used to absorb the sound waves propagating to boundaries, and a fixed constraint was applied at the bottom of the PMUT model. A measuring point was placed in front of the PMUT with a distance of 2 mm. The PVDF piezoelectric layer was driven by the electrical field applied between the top and bottom electrodes. We set the frequency range from 160 kHz to 260 kHz with a step size of 1 kHz. The sound pressure level (SPL) distribution after the simulation is shown in Figure 11a. The frequency response curve of the measuring point is plotted in Figure 11b. There is a response peak at approximately 203 kHz, which is consistent with the resonant frequency of the PMUT. This result also means that the PMUT has better performance in transmitting ultrasonic waves at this frequency. The simulation result of the sound pressure field is shown in Figure 11c. The peaks and troughs of ultrasonic waves propagating in the air domain can be seen from this result. We roughly fitted the troughs of the ultrasonic waves with arcs, and then calculated the distance between two arcs by comparing the numbers on the axis labels. The distance is approximately 1.65 mm. The wavelength (λ) can be expressed as λ =c/f where c is the sound speed in air medium, which is 340 m/s, and f is the frequency, which is 203 kHz. Through the equation, λ is calculated to be 1.67 mm, which is consistent with the measurement result. 

### 4.3. Frequency Response Analysis

The image of the final PMUT is shown in Figure 12. It was temporarily mounted on a glass sheet for easy testing. We characterized the frequency response of the device in air using a Laser Doppler Vibrometer (PSV-500F-B, Polytec China Ltd., Beijing, China). The device was excited by a periodic chirp signal in the frequency range from 130 kHz to 250 kHz with a voltage amplitude of 80 Vpp. The frequency response and the first vibration mode shape of PMUT are shown in Figure 13. From the results, we can see that there is a mild resonant peak at 198.37 kHz approximately, which is consistent with previous FEA simulations. The vibration region is located in the center of the grids (Figure 13a), corresponding to the position of the cavity. The vibration velocity in other regions is almost nil, indicating that the edge-fixed boundary condition is effective, which further illustrates the multilayered composite diaphragms are well bonded to the substrate without delamination. From Figure 13b, the frequency response curve has a feature with wide bandwidth, meaning that our PMUT exhibits a wide operational frequency range in applications.

### 4.4. Mechanical Characterizations

The effect of device bending on strain distribution was studied using FEA. Figure 14a shows that the maximum strain resides in the interface between the top electrode and PVDF film along the bending direction, due to the presence of the air cavity. The maximum strain increases exponentially with the decrease of bending radii (Figure 14b). Bending radii of 10, 8, 6 and 4 mm correspond to strain values of 0.11%, 0.17%, 0.31%, and 0.68%, respectively. Within the bending radius of 5 mm, the maximum strain is below 0.5%. There is a change point at the bending radius of 3 mm, corresponding to the maximum strain is 1.01%, which means that the performance of our device may degrade to some extent. Further improvements to the flexibility of our device include using a thinner substrate, optimizing the device shape to avoid stress concentration, and adding a top encapsulation layer according to the neutral plane theory [42].

Our PMUT can easily achieve conformal contact with flat, concave, and convex surfaces as shown in Figure 15a. The curvature radius of concave and convex models is 25 mm, and our PMUT can properly fit curved surfaces. During the bending process of the multilayered composite films, the stress on the outer surface is largest, resulting in the largest deformation. When the deformation exceeds the critical value, the films will break along the bending direction. The degree of deformation can be expressed by the relative bending radius: r/t. Where r is the bending radius, t is the thickness of the film. After that, the surface strain (ε) of top Ag electrode, PVDF film, and Kapton substrate can be calculated by the following equations: (3)$$\varepsilon_{\mathrm{Ag}}=\frac{t_{\mathrm{Ag}}}{2r+t_{\mathrm{Ag}}}$$
(4)$$\varepsilon_{\mathrm{PVDF}}=\frac{t_{\mathrm{PVDF}}}{2r+t_{\mathrm{PVDF}}}$$
(5)$$\varepsilon_{\mathrm{Kapton}}=\frac{t_{\mathrm{Kapton}}}{2r+t_{\mathrm{Kapton}}}$$
where $t_{\mathrm{Ag}}$, $t_{\mathrm{PVDF}}$, $t_{\mathrm{Kapton}}$ are the thicknesses of the top Ag electrode, PVDF film, and substrate, which are sequentially 10, 28 and 100 μm. r is the radius of the model, which is 17.5 mm. Using these functions, the surface strain values can be calculated: $\varepsilon_{\mathrm{Ag}}=0.028\%$, $\varepsilon_{\mathrm{PVDF}}=0.08\%$, and $\varepsilon_{\mathrm{Kapton}}=0.28\%$. The curved film will not crack under the condition of $\varepsilon \leq \delta_{\max}$, where $\delta_{\max}$ is the elongation of the material [43]. Because the dimension of film thickness is in micro-scale, ε is much less than $\delta_{\max}$, which will not affect the performance of the device. Under the premise of bending safely, a specially-made linear mobile platform was used to clamp and alternately bend or unbend the PMUT (Figure 15b). The testing results verified that our PMUT could endure biaxial bending forces and could survived consecutive mechanical deformations. We also pasted PMUT on the back of the hand and twined it around the wrist to demonstrate the suitability for skin applications, as shown in Figure 15c. These results confirmed that the flexibility and mechanical stability of our PMUT could satisfy the ordinary demands of wearable devices. 

### 4.5. Acoustic Characterizations

An acoustic transmit-receive system was set up in air ambience for the acoustic characterization of our PMUT. The schematic diagram of this system is shown in Figure 16. A sine burst with five cycles was produced by a function generator (AFG1062 Tektronix, Dongfang Zhongke Integrated Technology Co., Ltd., Beijing, China) and amplified by a power amplifier (75A250A, 10 kHz~250 MHz, Amplifier Research Corporation, Souderton, PA, USA) to drive the transmitter. The resulting signal from the receiver was amplified using Manually Controlled Ultrasonic Pulser-Receiver (5072PR, OLYMPUS, Tairu Electronic Technology Co., Ltd., Beijing, China) in receiving mode and finally displayed on an oscilloscope (DPO 2014, Tektronix, Dongfang Zhongke Integrated Technology Co., Ltd., Beijing, China). 

Firstly, our PMUT was applied as a transmitter and a 200 kHz commercial ultrasonic transducer (UT, JINCI Technology Co., Ltd., Shenzhen, China) was used as a receiver. The PMUT was tested in different states: a flat state, an up bending state, and a down bending state, which was shown in Figure 17a–c. The ultrasonic transducer was placed opposite to the PMUT and the distance between them was 20 mm. The PMUT was driven by an 80 Vpp burst with a frequency of 195 kHz. The transmitting signal was successfully caught by the ultrasonic transducer and was shown in Figure 17d. From the results, we can see that these curves overlap with each other, indicating that the performance of the flexible PMUT is well preserved under mechanical bending. The amplitude of the signal was 30 mVpp with a voltage gain of 40 dB, and its corresponding Fast Fourier Transform (FFT) exhibited a center frequency of approximately 195 kHz. 

Then the curved PMUT was used as a receiver and the ultrasonic transducer served as a transmitter. The PMUT was mounted on a cylinder with 25 mm curvature radius and wired to a BNC connector through the coaxial line (Figure 18a). The distance between them was also 20 mm. The ultrasonic transducer was excited by a 200 kHz burst with a voltage amplitude of 80 Vpp. The signal received by the PMUT was processed in MATLAB and a low-pass filter with the frequency of 300 kHz was applied to reduce the electrical noise. The multiple reflection of acoustic waves between the PMUT and ultrasonic transducer was captured as shown in Figure 18b. 

The time of flight (TOF) of the first two peaks could be measured from the time-domain curve, which was 121 μs. The distance between the PMUT and ultrasonic transducer can be computed from L=vt/2. Where L is the distance between two objects, v is the sound speed in air at room temperature, which is 340 **m/s**, and t is the time of flight. Using this function, the distance was estimated to be 20.57 mm, which was approximately consistent with the pre-set value. 

## 5. Conclusions

In this work, a flexible PMUT operating in air was successfully designed, fabricated and characterized. Finite element analysis was employed to predict the resonant frequency of composite diaphragms and optimize the structure. A low temperature adhesive bonding technique which aims to minimize the fabrication steps and reduce the costs was used to manufacture the PMUT stably and effectively. The resulting device based on flexural vibration mode has a center resonant frequency of 198 kHz with a wide operational bandwidth. Our device has good conformal contacting with flat, concave and convex surfaces, and survives continuous stretched and compressive bending forces. Furthermore, an acoustic transmit-receive system has been established to demonstrate the acoustic characterization of our device. These experimental results confirm that the proposed PMUT has the potential to be integrated with intelligent devices and wearable electronics.

## Figures and Tables

**Figure 1 sensors-20-03333-f001:**
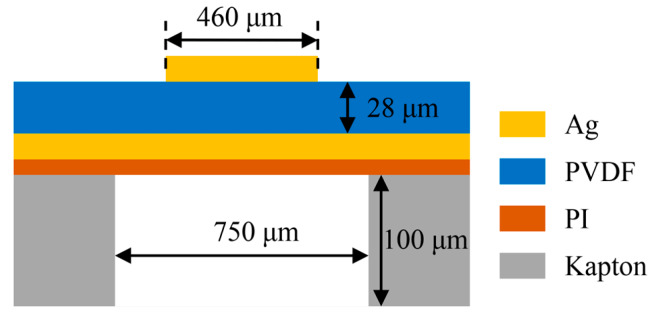
The schematic cross-section of the PMUT.

**Figure 2 sensors-20-03333-f002:**
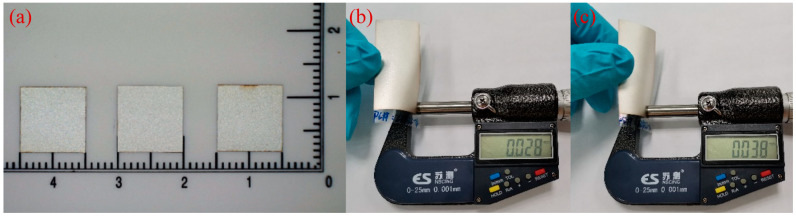
The images of square PVDF films (**a**), and the thickness measurements of uncoated PVDF film (**b**), and one-side Ag coated PVDF film (**c**).

**Figure 3 sensors-20-03333-f003:**
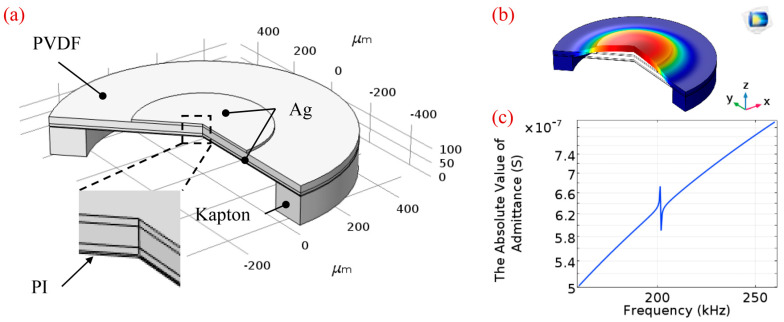
The FEA of the PMUT using COMSOL Multiphysics 5.3. (**a**) Simulation model; (**b**) vibration mode shape at the resonant frequency of 202.58 kHz; (**c**) the chart of the admittance.

**Figure 4 sensors-20-03333-f004:**
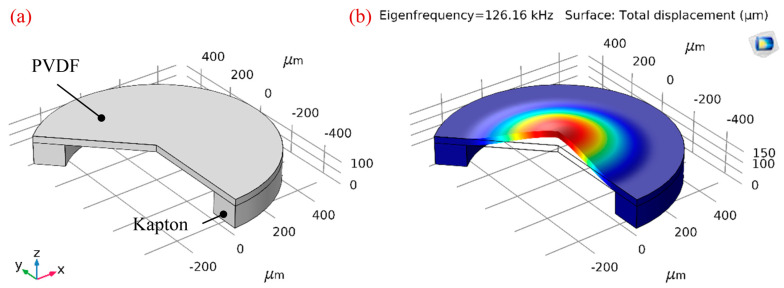
The simulation results of PVDF based monolayer structure. (**a**) Simulation model built in COMSOL Multiphysics 5.3; (**b**) vibration mode shape at the resonant frequency of 126.16 kHz.

**Figure 5 sensors-20-03333-f005:**
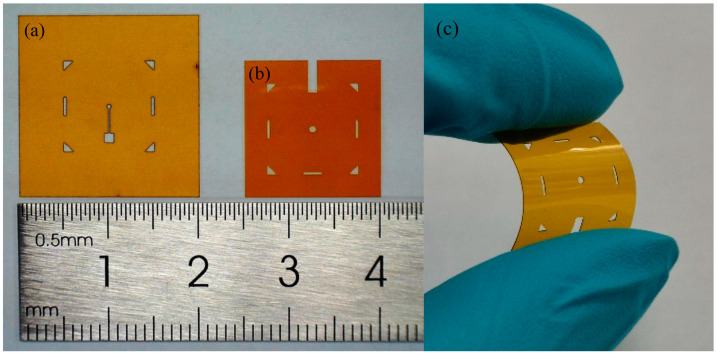
The optical image of commercial Kapton handled by laser precision machining system. (**a**) A 10 μm thick Kapton tape used for printing mask; (**b**) a 100 μm thick Kapton sheet served as a flexible substrate with a through-hole; (**c**) a flexible Kapton substrate.

**Figure 6 sensors-20-03333-f006:**
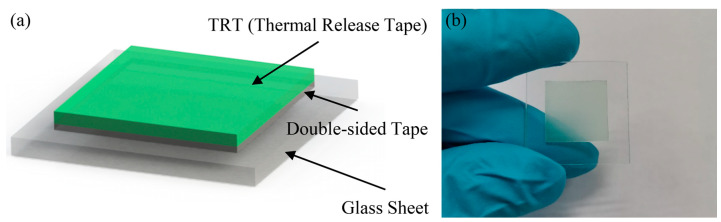
The schematic diagram (**a**) and the optical image (**b**) of the temporary carrier.

**Figure 7 sensors-20-03333-f007:**
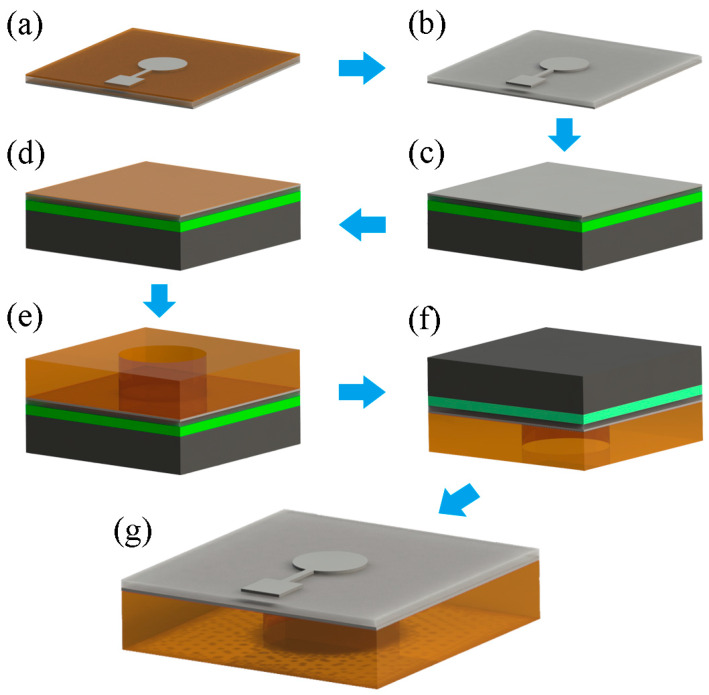
The schematic illustration of the low temperature adhesive bonding fabrication process. (**a**) Pattern top Ag electrode, (**b**) Ag coated PVDF film, (**c**) Ag coated PVDF film pasted on the temporary carrier upside down, (**d**) spin coating a 4 μm thick PI, (**e**) device assembly, (**f**) solidify the PI and deactivate the TRT, (**g**) the final device.

**Figure 8 sensors-20-03333-f008:**
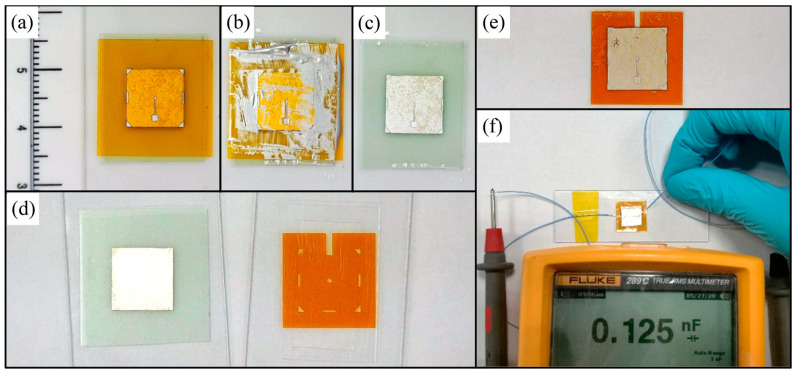
The additional images of some steps in fabrication process. (**a**) The printing mask pasted on one-side Ag coated PVDF film. (**b**) Transferring top electrode pattern from the printing mask to PVDF film. (**c**) Ag patterned PVDF film after peeling off the printing mask. (**d**) The PVDF film and the substrate prepared for adhesive bonding process. (**e**) The final PMUT through the low temperature adhesive bonding fabrication process. (**f**) A simple capacitance testing for fabricated device.

**Figure 9 sensors-20-03333-f009:**
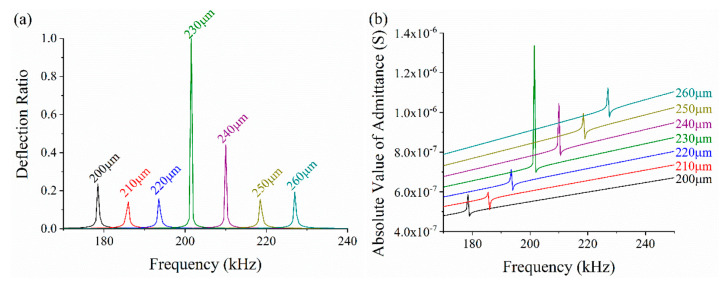
The FEA simulation results of parametric sweeping. (**a**) Deflection displacement under conditions of various radii, (**b**) the absolute value of admittance.

**Figure 10 sensors-20-03333-f010:**
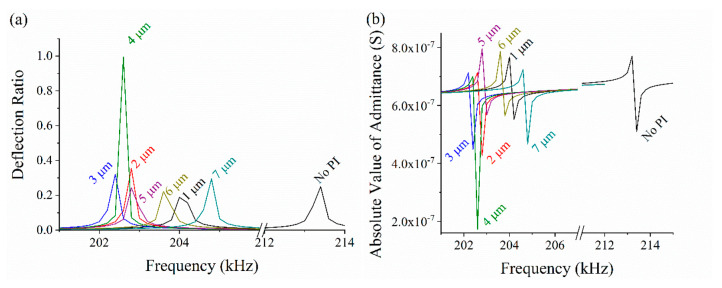
(**a**) Deflection displacement of the measuring point. (**b**) The chart of the admittance.

**Figure 11 sensors-20-03333-f011:**
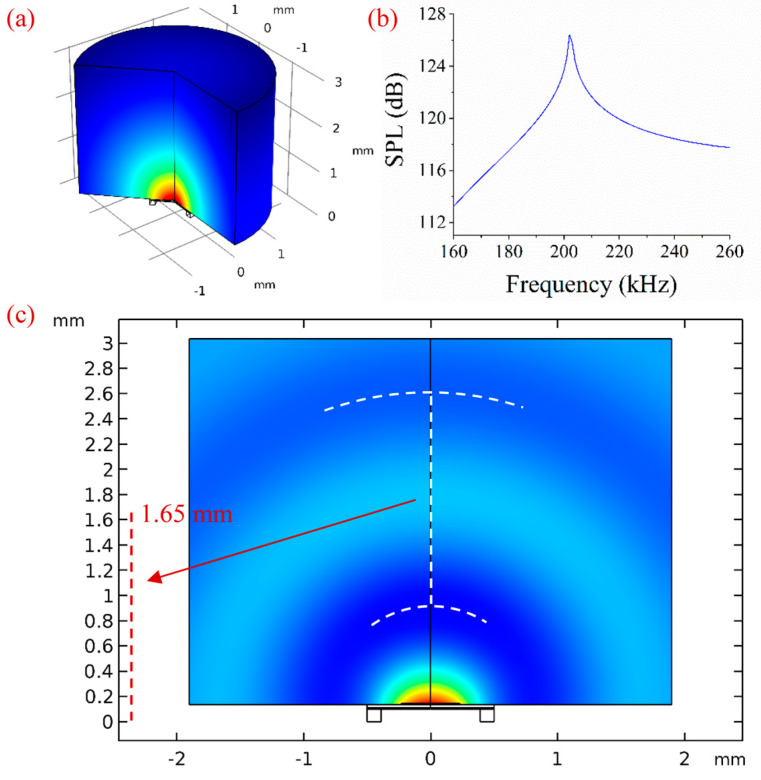
(**a**) The sound pressure level (SPL) in air domain. (**b**) The frequency response curve. (**c**) Simulation of the sound pressure field.

**Figure 12 sensors-20-03333-f012:**
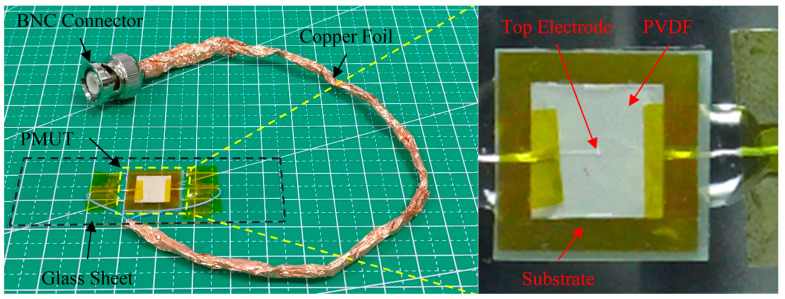
The optical image of PMUT mounted on a glass sheet.

**Figure 13 sensors-20-03333-f013:**
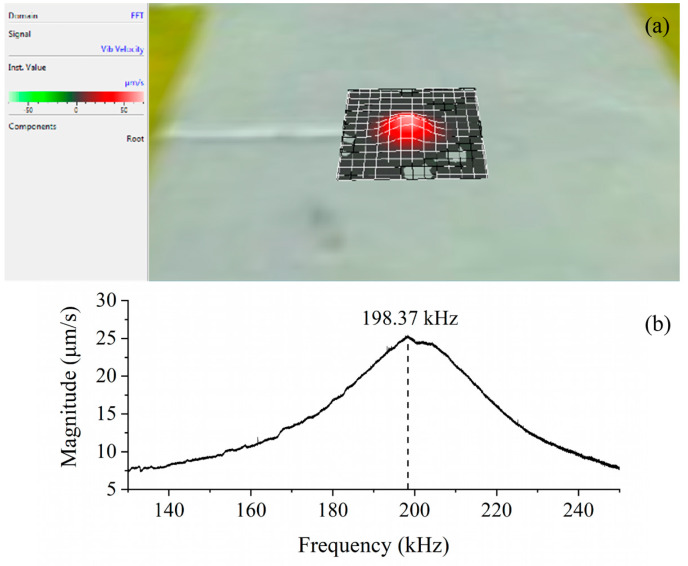
The frequency response analysis of proposed PMUT. (**a**) The flexural vibration mode shape; (**b**) the frequency response curve.

**Figure 14 sensors-20-03333-f014:**
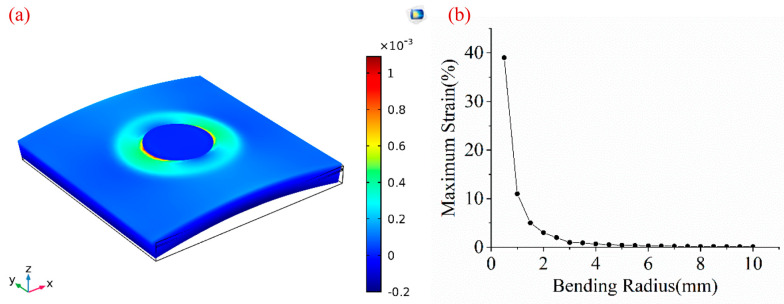
(**a**) The chart of maximum strain solved by FEA. (**b**) Correlation of the bending radius versus maximum strain in a device membrane.

**Figure 15 sensors-20-03333-f015:**
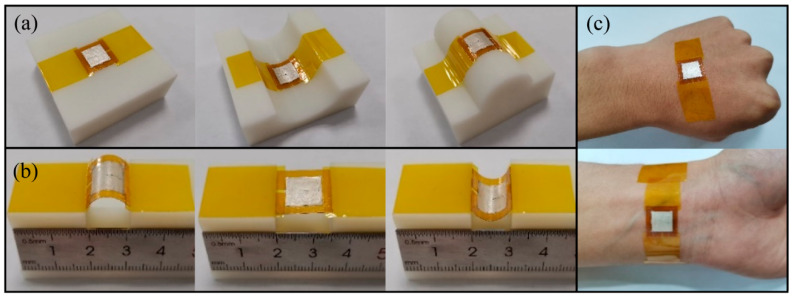
The optical images of PMUT (**a**) bonded on flat (left), concave (middle), and convex (right) surfaces, (**b**) clamped on a custom-designed linear mobile platform in up-bending (left), flat (middle) and down-bending (right) configurations, and (**c**) contacted to the skins.

**Figure 16 sensors-20-03333-f016:**
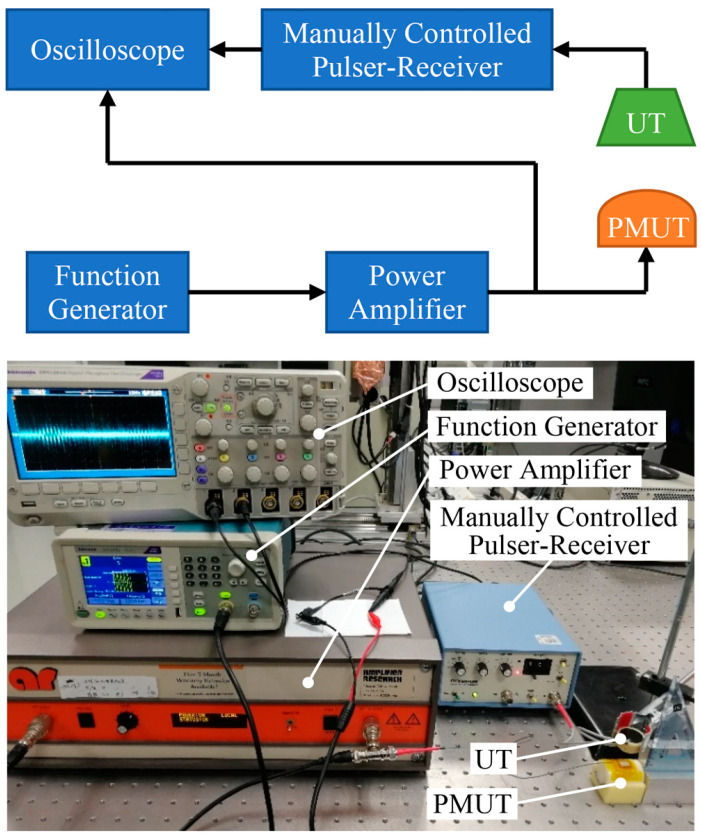
Acoustic transmit-receive system.

**Figure 17 sensors-20-03333-f017:**
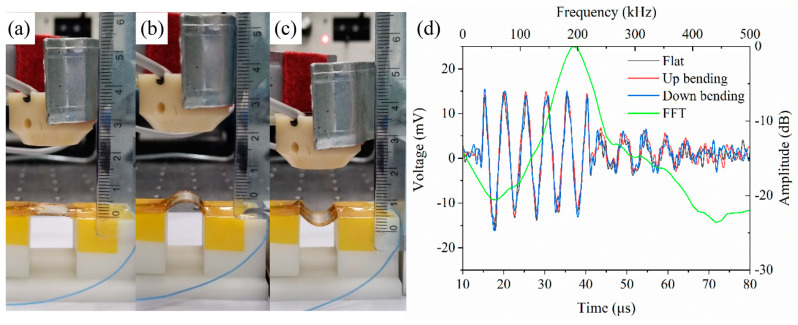
The PMUT served as a transmitter in flat state (**a**), up bending state (**b**), and down bending state (**c**). (**d**) The receiving signals and the FFT spectrum.

**Figure 18 sensors-20-03333-f018:**
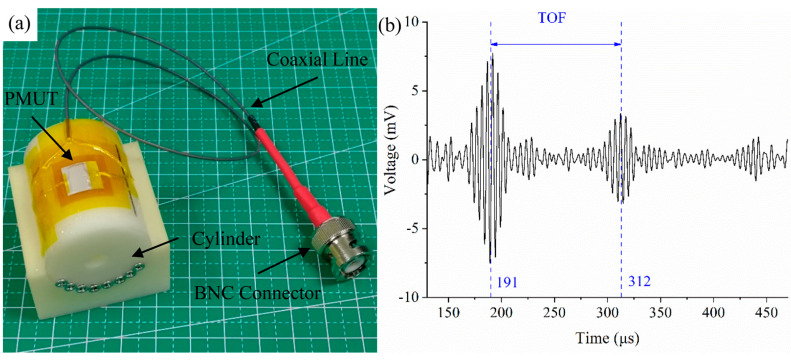
(**a**) PMUT mounted on a cylinder with 30 mm curvature radius. (**b**) Time of flight (TOF) measurement using PMUT on curved surface.

**Table 1 sensors-20-03333-t001:** The material properties used in FEA.

Material	Mass Density (kg/m^3^)	Young’s Modulus (GPa)	Poisson’s Ratio
PI	1300	3.1	0.37
Ag	10,500	83	0.38

**Table 2 sensors-20-03333-t002:** Laser parameters used for ablating the pattern.

Material	Frequency (kHz)	Power (W)	Speed (mm/s)	Repetition
10 μm Kapton tape	70	0.7	300 mm/s	25
100 μm Kapton sheet	70	1	300 mm/s	35

**Table 3 sensors-20-03333-t003:** The resonant frequencies with different radii of top electrodes.

**Radius (μm)**	200	210	220	230	240	250	260
**Resonance Frequency (kHz)**	179.61	186.82	194.67	202.58	211.06	219.67	228.13
**Increasing Value (kHz)**	0	7.21	7.85	7.91	8.48	8.61	8.46

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
