# Peer review of "Low Temperature Adhesive Bonding-Based Fabrication of an Air-Borne Flexible Piezoelectric Micromachined Ultrasonic Transducer"

_sensors, 2020, doi:10.3390/s20113333_

Round 1
Reviewer 1 Report
In the paper, a particular kind of ultrasonic piezoelectric transducer is proposed and studied. The authors perform both numerical FEA and experimental tests to illustrate the features of the proposed transducer.
The paper is sufficiently clear and could be published but after some changes. Please see the points below.
Please check the text because there are many typos.
In the title please check "Air-bone". Are you sure of it or do you mean "airborne"?
Please check carefully the unit of measures. They are throughout the text very badly typed. Especially, "KHz" must be corrected in all places. Indeed, this means "kelvin herz" but of course you mean "kHz" kilohertz.
The same goes for "Kg" kelvingram that must be "kg" kilogram on page 3 line 104 as well as in table 1.
At page 1 line 32, please correct "which" with the word "that".
In figure 1, please use micrometer instead of the mysterious unit "um".
At page 3 line 100 what is the density? Perhaps do you mean "mass density"? Please correct also Table 1.
At page 3 line 101, please correct "the Young’s modulus" and "the Poisson’s ratio", they must be "Young’s modulus" and "Poisson’s ratio" or "the Young modulus" and "the Poisson ratio".
At page 3 line 101, please change the name of Poisson's ratio, the choice of "mu" is very unfair since with "mu" is denoted very often the shear elastic modulus. The reviewer suggests using the more common Greek letter "nu".
The equations 1, 2, 3 are redundant. Equation 3 does not add any new information.
At page 4 lines 126-127, the claim "So h has a greater impact on ? than the square root of ? over ?" is not clear. Please explain what do you intend? The function f is proportional to both these two quantities. Therefore the impact is exactly the same. Maybe, you mean that changing the thickness is more simple than changing the material. Please check this point.
The caption of figure 7 must follow immediately the figure.
Please correct the strange unit of measure of figure 16. It is not clear the meaning of "us". Is it "microsecond"?
In the introduction, the authors should do a more comprehensive literature review. For example, they consider the wearability as a useful feature, so they can also talk about piezoelectric or flexoelectric transducers made by fibers with a proper arrangement of them to improve the wearability (see, e.g. [1,2]).
[1] Giorgio, I., Della Corte, A., dell'Isola, F. and Steigmann, D. J. (2016) Buckling modes in pantographic lattices. Comptes Rendus - Mecanique, 344: 487-501.
[2] Eremeyev, V. A., Ganghoffer, J. F., Konopińska-Zmysłowska, V., & Uglov, N. S. (2020). Flexoelectricity and apparent piezoelectricity of a pantographic micro-bar. International Journal of Engineering Science, 149, 103213.
Reviewer 2 Report
Comments to the Authors,
1. Comment #1: I would like to recommend modifying the title because the title at first glance seemed a bit vague, whether the paper focused on low-temperature adhesive bonding or a simply fabricated flexible PMUT. I would suggest some examples as “ Low-temperature adhesive bonding-based fabrication of ….” or “Fabrication of …. Transducer by low-temperature adhesive bonding”.
2. Comment #2: When the flexible devices stuck on different surfaces, it can be pre-stressed from the initial condition. The initial membrane of the PMUT could bend up or down when it confronts compressive or tensile stress, the simulation did not take account for those conditions. It would be better to see that additional simulation results of the pre-stressed conditions can be obtained and be included in the paper. Also, no consolidated experimental data was provided for this. This should be very important to provide these data since the flexible PMUT can be differentiated from other normal PMUT. This should be included in the discussion section.
3. Comment #3: The authors have to make clear about what kind of equipment they used in the experiments. The authors should provide all the specific models or materials. Primarily, there were no details on adhesive bonding fabrication and their materials to make flexible PMUT. It is necessary to provide readers the material information and fabrication process in detail of the thermal release tape. Also, what is the specification of the “manually controlled pulser-receiver” when the authors used in acoustic characterization?
4. Comment #4: Clarify the evidence of no residual adhesive remained on the PMUT surface after deactivating and separating the thermal release tape. Also, is there any performance degradation of the flexible PMUT with some residues left from the thermal release tape?
5. Comment #5: In line 97–102 and Line 126–130, for the calculation of the resonance frequency, h is defined as the thickness of the membrane first in Line 100, and this is changed in Line 127 to represent the thickness of the whole diaphragms for explaining the reason of different resonance frequencies between theoretical calculation and simulation. This can be a little confusing and thus needs clarification. Adding a simulation result on the resonance frequency when there is only the membrane could be beneficial to readers to show correlations between calculation and simulation as well as differences among the diaphragm structures.
6. Comment #6: In line 140, the author claimed the laser precision machining is timesaving, low-cost, and can offer a high yield compared with micromachining fabrication techniques. However, a commonly massive production with standard photolithography, etching, and evaporation can drop the price more than the laser precision machining. Also, the minimum pattern size achieved using a laser might be limited by its beam size, which can be smaller than that of conventional microfabrication techniques. In this regard, it would be helpful to provide more information on the spatial resolution (the minimum feature size) of the laser system used in this paper, for future uses of this technique to fabricate smaller PMUTs.
7. Comment #7: In Section 4.1, for clarification, the authors should specify fixed parameters in each optimization process, e.g., the thickness of a PI layer during the electrode diameter optimization. Also, no units were notified on the y-axes in figure 8 and 9.
8. Comment #8: In Section 4.2, the authors wrote that the calculation and experiment of continuous deformation evaluated the mechanical stability of the fabricated PMUT. It would be more persuasive, providing some more results that show their mechanical stability by changing the resonance frequency as a function of bending cycles.
9. Comment #9: There are several errors in a few sentences that need to be taken care of. For example, “Air-bone” in Line 3 should be “Air-borne”, and “Figure 4” in Line 161 should be “Figure 6.”
Round 2
Reviewer 2 Report
In the revised edition of the manuscript, the authors answered all of my questions and comments in detail. To a great extent, they fulfilled all of my doubts in the paper. This paper would be accepted after minor revision to correct a few errors or text editing.
Author Response
Dear Editor and Reviewer:
We would like to thank you for giving us a chance to minor revise the paper. The manuscript “Low-Temperature Adhesive Bonding-based Fabrication of an Air-bone Flexible Piezoelectric Micromachined Ultrasonic Transducer” (sensors-819251), authored by Wei Liu and Dawei Wu, has been revised carefully. According to the reviewer’s comments and suggestions, we have checked the number of each Figure, improved the structure of some sentences, and corrected grammatical errors. Revised portions are printed in red in the manuscript.
This manuscript is a resubmission of an earlier submission. The following is a list of the peer review reports and author responses from that submission.